# Event Factuality Identification via Deep Neural Networks

## Abstract

Event factuality identification plays an important role in deep NLP applications. In this paper, we propose a deep learning framework for this task which first extracts essential information from raw texts as the inputs and then identifies the factuality of events via a deep neural network with a proper combination of Bidirectional Long Short-Term Memory (BiLSTM) neural network and Convolutional Neural Network (CNN). The experimental results on FactBank show that our framework significantly outperforms several state-of-the-art baselines.

## 1 Introduction

Event factuality is the information expressing the commitment of relevant sources towards the factual nature of events. That is, event factuality conveys whether an event is characterized as a fact, a certain possibility, or an impossible situation, and thus is helpful for various NLP applications, e.g., opinion detection (Wiebe et al., 2005), QA (Pustejovsky et al., 2005), textual entailment (de Marneffe et al., 2006; Hickl and Bensley, 2007), rumor identification (Qazvinian et al., 2011).

In principle, the value of event factuality is related to some elements, e.g., predicates, speculative and negative cues. Consider following sentences as examples[1]:

(S1) _McCulley, a famous economist, **doubts** that the tax rate will **increase** soon._

(S2) _He **knows** they are not able to **go** to the village due to the flood._

In sentence S1, the event **increase** is a possibility according to the predicate **doubts**, while in

S2, the event **go** is regarded as an impossible situation due to the negation word _not_. From these sentences, we can learn that predicates and cues can determine the event factuality to a great degree. Such event factuality provides the way to distinguish fact events from speculative and negative ones.

Traditional methods for event factuality identification usually employed either hand-crafted rules (Saurí, 2008; Saurí and Pustejovsky, 2012) relying on expert knowledge, or features (Prabhakaran et al., 2010; de Marneffe et al., 2012) relying on various kinds of annotated information, such as source introducing predicates, predicate classes, speculative and negative cues. Due to the recent success of deep learning on various NLP tasks in learning useful representations from ordinary sentences (Socher et al., 2012; Zeng et al., 2014; Cheng et al., 2016) and syntactic paths (Xu et al., 2015a,b; Roth and Lapata, 2016) and the attention mechanism on capturing the focus change in sequence modeling (Chen et al., 2016; Wang et al., 2016; Zhou et al., 2016), this paper proposes a framework to identify event factuality in raw texts with neural networks. Our main contributions can be summarized as follows:

1) the proposal of a two-step supervised framework for identifying event factuality in raw texts.

2) the utilization of an attention-based CNN to detect source introducing predicates (SIPs), the most important factors to identify event factuality.

3) the proposal of an attention-based deep neural network model with a proper combination of BiLSTM and CNN to identify the factuality of events.

## 2 Background

This section presents the concepts and basic factors of event factuality.

---

[1] In this paper, events are in **bold** and sources are _underlined_ in example sentences.

## 2.1 Event Factuality

Factuality can be characterized by the combination of two dimensions: epistemic modality and polarity (Saurí, 2008). While modality conveys the certainty degree of events, such as *certain* (CT), *probable* (PR), *possible* (PS) and *underspecified* (U), polarity expresses whether the event happened, including *positive* (+), *negative* (-) and *underspecified* (u). Table 1 shows various values of event factuality. For example, CT+ means it is certain that the event happened, while PR- denotes it is probable that the event did not happen. Notice that some values are non-application (NA) grammatically, e.g., PRu, PSu, U+/-.

|     | +     | -     | u     |
|-----|-------|-------|-------|
| CT  | CT+   | CT-   | CTu   |
| PR  | PR+   | PR-   | (NA)  |
| PS  | PS+   | PS-   | (NA)  |
| U   | (NA)  | (NA)  | Uu    |

Table 1: Various values of event factuality.

## 2.2 Basic Factors

Considering that FactBank is built on top of TimeBank (Pustejovsky et al., 2003b), we adopt the definition of events proposed by TimeML (Pustejovsky et al., 2003a) from a more grammatical and generalized perspective. We consider the events which can be critical for computing the factuality. Such events can be detected by feature-based methods (Chambers, 2013).

In the literature, three kinds of factors play critical roles in event factuality identification: Source Introducing Predicate (SIP), source and cue.

**SIP**. Source Introducing Predicates (SIPs) are events that can not only introduce additional sources to assess the factuality of embedded events, but also influence their factuality. For example, the SIP *doubts* introduces *McCulley* as a new source in S1, and *McCulley* evaluates event *increase* as a possibility according to *doubts*.

**Source**. By default, *AUTHOR* is considered the only source if the sentence contains no SIPs. Further sources can be incorporated by SIPs. In S1, the event *increase* has two sources: *AUTHOR* and *McCulley*. The factuality value is not the inherent property of the event but always relevant to the sources. *McCulley* commits to the event *increase* as a possibility, while *AUTHOR* remains uncommitted. Strictly speaking, we as readers know about *McCulley*'s perspective only according to *AUTHOR*. Hence, we appeal to the notion of the nested source structure (Wiebe et al., 2005) and represent the sources in the chain form: *McCulley_AUTHOR*, which means *McCulley* is **Embedded** in *AUTHOR*. Only *AUTHOR* can be displayed by a simple source.

**Cues** are words that have speculative or negative meanings. The modality of a Non-Uu event is PR/PS if this event is modified by a PR/PS cue, while the polarity of a Non-Uu event is negative if this event is modified by a negative (NEG) cue. So cues are key signals to identify the event factuality. For instance, the factuality of event **go** in S1 is CT- due to the NEG cue *not*.

## 3 Baseline

As a baseline, we present a two-step framework shown in Figure 1 to identify event factuality in raw texts. Firstly, various useful factors, such as SIPs, sources, and cues are extracted from raw texts. Then, a classification algorithm is employed to identify event factuality.

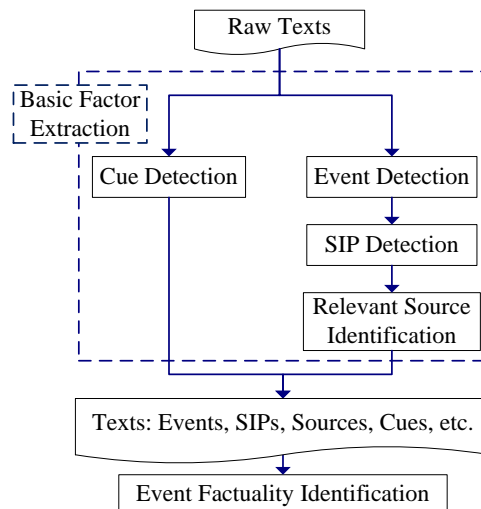

Figure 1: The framework of event factuality identification task.

## 3.1 Event and Cue Detection

For event detection task, we only consider nouns, verbs, adjectives as the candidate tokens to reduce the complexity, and we utilize a maximum entropy classification model with the features used by Chambers (2013).

Øvrelid et al. (2010) and Velldal et al. (2012) concluded that syntactic features are not necessary and lexical sequence-oriented n-gram features can achieve excellent performances on cue detection

task. Hence, we employ the *lexical features* developed by Velldal et al. (2012) to classify each token into *PR cue*, *PS cue*, *Neg cue*, or *not cue*.

## 3.2 SIP Detection

Whether an event is a SIP has relation to its semantic information. For instance, verbs *state, say, tell* are usually used to express opinions which are promoted by certain participants of events, so they can all introduce additional sources. Therefore, they are in the same synonym set {*state, say, tell*} and share the same hypernym in WordNet[2]. Hence, we consider *token*, *part-of-speech (POS)* and *hypernym* of the token as **lexical level features**, and concatenate them into the vector *l*.

According to the definition of SIPs, if there is no event in the syntactic scope of a word, this word is not a SIP. For example:

(S3) *Tom **suggests** we should have a rest now*.

In S3, although the event ***suggests*** can introduce a new source *Tom* for the clause "*we should have a rest now*", however, this clause expresses a suggestion and is not an event. So the event ***suggests*** is NOT a SIP in S3. This example demonstrates the structure of a sentence is also able to determine whether a token is a SIP. Therefore, we also consider the **sentence level feature**. Instead of original sentences, we propose the following **Pruned Sentence** structure:

**Pruned Sentence (PSen)**: If a clause of the candidate token contains events, this clause is replaced by the tag $\langle event \rangle$; Nouns, pronouns and the current candidate token are unchanged, while other tokens are replaced by the tag $\langle O \rangle$.

We argue that PSen can characterize clearly whether there are events in the clause of the token. Besides, in PSen only the candidate token and the tokens that are possible to be new sources are reserved, and effects of other tokens are omitted.

For example, in S4, the current candidate event ***says*** governs the clause "*the manager will **attend** a meeting later*", which contains the event ***attend***. Hence this clause is replaced by the tag $\langle event \rangle$. While the tokens which are impossible to be new sources are replaced by $\langle O \rangle$. Sentence S5 is the PSen structure of S4.

(S4) *Tom, who is the secretary of the manager, **says** the manager will **attend** a meeting later* .

(S5) *Tom* $\langle O \rangle$ *who* $\langle O \rangle$ $\langle O \rangle$ *secretary* $\langle O \rangle$ $\langle O \rangle$ *manager* $\langle O \rangle$ ***says*** $\langle event \rangle$ $\langle O \rangle$

---
[2]http://wordnet.princeton.edu

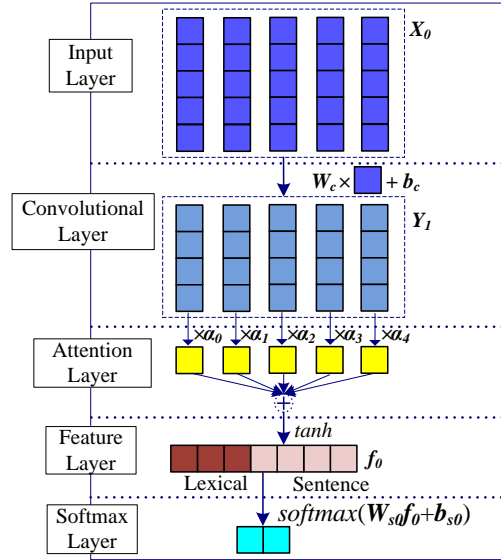

Figure 2: The CNN for SIP detection.

Considering that PSen is a simplified and pruned structure, we extract sentence level features through an attention-based CNN illustrated in Figure 2 instead of RNN model. A pruned sentence $S = (t_0, t_1, \ldots, t_{n-1})$ is transferred to a matrix $X_0 \in \mathbb{R}^{d_0 \times n}$ according to pre-trained word embeddings. Our CNN is computed as follows:

$$Y_1 = W_c X_0 + b_c \qquad (1)$$
$$Y_m = \tanh(Y_1) \qquad (2)$$
$$\alpha = softmax(v_c^T Y_m) \qquad (3)$$
$$c = \tanh(Y_1 \alpha^T) \qquad (4)$$

where $W_c \in \mathbb{R}^{n_c \times d_0}$, $b_c, v_c, c \in \mathbb{R}^{n_c}$. The lexical feature $l$ includes the embeddings of the *token*, the *POS tag* and the *hypernym* with the dimension of $d_0$, $d_{pos}$ and $d_{hyp}$, respectively. We concatenate $c$ and $l$ into $f_0 = [l^T, c^T]^T$ for each token. Finally, $f_0$ is fed into the softmax layer:

$$o = softmax(W_{s0} f_0 + b_{s0}) \qquad (5)$$

where $o \in \mathbb{R}^2$. To train the network, we exploit the following objective function:

$$J(\theta) = -\frac{1}{m} \sum_{i=0}^{m-1} \log p(y^{(i)}|x^{(i)}, \theta) + \frac{\lambda}{2} \|\theta\|^2 \quad (6)$$

where $p(y^{(i)}|x^{(i)}, \theta)$ is the confidence score of the golden label $y^{(i)}$ of the training instance $x^{(i)}$, $m$ is the number of the training instances, $\lambda$ is the regularization coefficient and $\theta$ is the parameter set.

The grammatical subjects of SIPs are chosen as the **New Sources** introduced by corresponding

SIPs. Since we have identified events and SIPs, we identify relevant sources $RS$, which is initialized as $RS = \{AUTHOR\}$, for each event according to the definition. When traversing from the root of the dependency parse tree of the sentence to the current event, $RS$ is updated as:

$$RS_n = RS_{n-1} \cup \{ns\_s\} \qquad (7)$$

where $ns$ is a new source introduced by the corresponding SIP and $s \in RS_{n-1}$. This is a recursive algorithm defined by Saurí (2008).

## 4 Deep Neural Network Framework

This section describes our neural networks for event factuality identification in details shown in Figure 3. The inputs are developed according to the outputs of Section 3. We learn effective representations from **Shortest Dependency Paths (SDP)** by means of neural networks with attention. Different from previous studies, our model combines both BiLSTM and CNN properly which can learn meaningful features from SDPs and words, respectively.

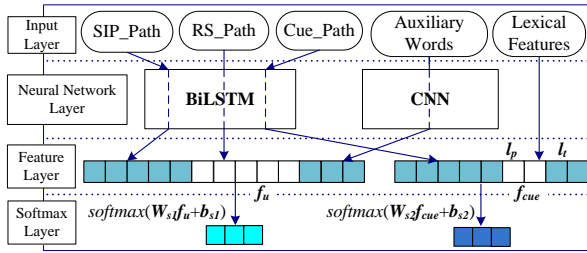

Figure 3: The architecture of our neural networks for event factuality identification.

Our model has two outputs: one represents whether the event is Uu, Non-Uu or other, and the other shows whether the event is modified by the cue and then Non-Uu events are further classified into CT+/-, PR+/-, PS+/-. We have two main reasons for the design of the two outputs: 1) We can identify negative and speculative values (e.g., CT-, PR+/-, PS+/-) more precisely with the assistance of cues, and 2) this design can tackle the imbalance among instances, because the negative and speculative factuality values are usually in the minority. Finally, the event factuality is determined by these two outputs directly.

### 4.1 Inputs

We develop the following inputs for our neural network model:

**Relevant Source Path (RS_Path)**: The SDP from the root of the dependency tree to the relevant sources of the event. Notice that this path contains all the sources in the chain form.

**SIP_Path**: The SDP from the SIP that introduces the current source to the event.

**Cue_Path**: The SDP from the cue to the event.

In addition, we consider the **Lexical Features** to judge whether the event is modified by the cue: 1) **Relative Position** is defined as the distance from the cue to the event in the sentence, and is mapped into the vector $\boldsymbol{l}_p$ with the dimension $d_p$; 2) **Type of Cue** includes PR, PS and NEG and is mapped into the vector $\boldsymbol{l}_t$ with the dimension $d_t$.

**Auxiliary Words (Aux_Words)** include auxiliary and marker words that share the dependency relations *aux* or *mark* with the current event. Here, a marker is the word introducing a finite clause subordinate to the event. We argue that Aux_Words can describe the syntactic structures of sentences.

An example sentence and its inputs for our neural networks are shown in Figure 4.

### 4.2 LSTM with Attention

Traditional RNNs have problems called vanishing gradients during the gradient back-propagation phase. Overcoming these problems is the motivation behind the LSTM model (Hochreiter and Schmidhuber, 1997) which introduces a memory cell including an input gate $\boldsymbol{i}_t$, a neuron with a self-recurrent connection $\boldsymbol{c}_t$, a forget gate $\boldsymbol{f}_t$ and an output gate $\boldsymbol{o}_t$. The values of components depend on the previous state $\boldsymbol{h}_{t-1}$ and the current input $\boldsymbol{x}_t$. Formally, LSTM is computed according to the following equations, where $\boldsymbol{x}_t \in \boldsymbol{X}$, $\boldsymbol{X} \in \mathbb{R}^{d_0 \times n}$ is the matrix of the SDP:

$$\boldsymbol{i}_t = \sigma(\boldsymbol{W}_i \boldsymbol{x}_t + \boldsymbol{U}_i \boldsymbol{h}_{t-1} + \boldsymbol{b}_i) \qquad (8)$$
$$\boldsymbol{f}_t = \sigma(\boldsymbol{W}_f \boldsymbol{x}_t + \boldsymbol{U}_f \boldsymbol{h}_{t-1} + \boldsymbol{b}_f) \qquad (9)$$
$$\boldsymbol{o}_t = \sigma(\boldsymbol{W}_o \boldsymbol{x}_t + \boldsymbol{U}_o \boldsymbol{h}_{t-1} + \boldsymbol{b}_o) \qquad (10)$$
$$\tilde{\boldsymbol{c}}_t = \tanh(\boldsymbol{W}_c \boldsymbol{x}_t + \boldsymbol{U}_c \boldsymbol{h}_{t-1} + \boldsymbol{b}_c) \qquad (11)$$
$$\boldsymbol{c}_t = \boldsymbol{f}_t \odot \boldsymbol{c}_{t-1} + \boldsymbol{i}_t \odot \tilde{\boldsymbol{c}}_t \qquad (12)$$
$$\boldsymbol{h}_t = \boldsymbol{o}_t \odot \tanh(\boldsymbol{c}_t) \qquad (13)$$

where $\odot$ is the element-wise multiplication and $\sigma$ is the sigmoid function. The dimension of hidden units in LSTM is set as $d_0$. We can obtain $\boldsymbol{H} \in \mathbb{R}^{d_0 \times n}$ via the LSTM layer, where $\boldsymbol{H} = (\boldsymbol{h}_0, \boldsymbol{h}_1, \ldots, \boldsymbol{h}_{n-1})$, $\boldsymbol{h}_t \in \mathbb{R}^{d_0}(0 \leq t \leq n-1)$.

Many sequence modeling tasks are beneficial from the access to the future as well as past con-

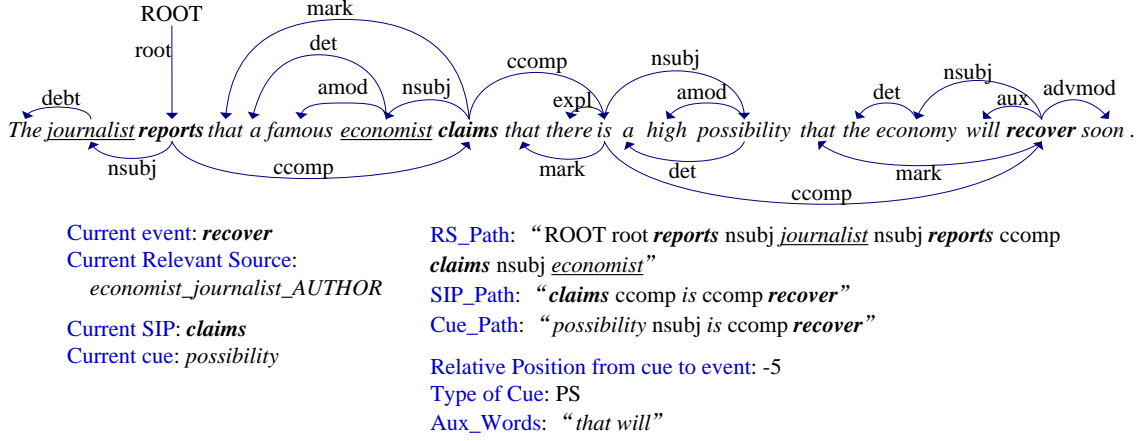

Current event: *recover*
Current Relevant Source:
 *economist_journalist_AUTHOR*

Current SIP: *claims*
Current cue: *possibility*

RS_Path: "ROOT root *reports* nsubj *journalist* nsubj *reports* ccomp *claims* nsubj *economist*"
SIP_Path: "*claims* ccomp *is* ccomp *recover*"
Cue_Path: "*possibility* nsubj *is* ccomp *recover*"

Relative Position from cue to event: -5
Type of Cue: PS
Aux_Words: "*that will*"

Figure 4: An example sentence and its inputs.

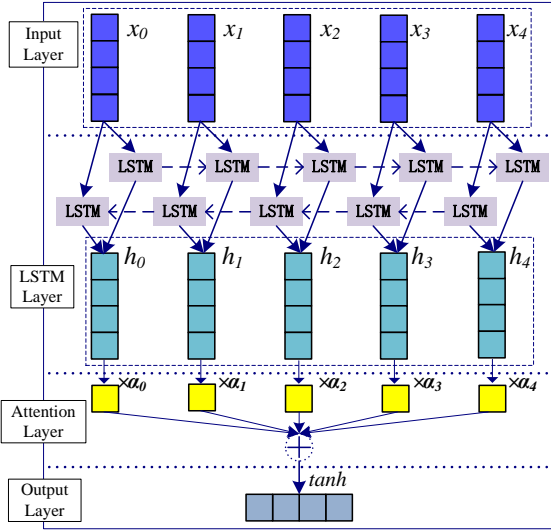

Figure 5: The bidirectional LSTM network with attention mechanism.

text. Therefore, to model the representations of syntactic paths, we utilize the bidirectional LSTM network shown in Figure 5 which processes the syntactic path in both directions. It produces the forward hidden sequence $\vec{H}$, the backward hidden sequence $\overleftarrow{H}$ and the output sequence $H_p$:

$$H_p = \vec{H} + \overleftarrow{H} \quad (14)$$

To capture the most important information from the syntactic path effectively, we adopt the attention mechanism and get the output $h_p$:

$$M = \tanh(H_p) \quad (15)$$
$$\alpha = softmax(v^T M) \quad (16)$$
$$h_p = \tanh(H_p \alpha^T) \quad (17)$$

### 4.3 Outputs

The feature representations $h_{sp}$, $h_{rp}$ and $h_{cp}$ are extracted from SIP_Path, RS_Path and Cue_Path,

respectively. Noticing that the auxiliary words are just a collection of tokens instead of a sequence with specific meanings, we employ the CNN in Section 3, i.e., Equation (1) to (4), to learn the representations of auxiliary words:

$$f_w = CNN(X_{aux\_words}) \quad (18)$$

We concatenate $h_{sp}$, $h_{rp}$ and $f_w$ into $f_u$ to judge whether the event is Uu, Non-Uu or other:

$$f_u = [h_{sp}^T, h_{rp}^T, f_w^T]^T \quad (19)$$

To determine whether a Non-Uu event is governed by a cue, we consider not only $h_{cp}$ but also the lexical features $l_p$ (Relative Position) and $l_t$ (Type of Cue) described above:

$$f_{cue} = [l_p^T, l_t^T, h_{cp}^T]^T \quad (20)$$

Finally, these feature representations are fed into the softmax layer:

$$o_1 = softmax(W_{s1} f_u + b_{s1}) \quad (21)$$
$$o_2 = softmax(W_{s2} f_{cue} + b_{s2}) \quad (22)$$

where $W_{s1}$, $b_{s1}$, $W_{s2}$, $b_{s2}$ are the parameters. The dimension of $o_1$ is 3, which is equal to the number of labels representing whether the event is Uu, Non-Uu or other (label $y_1$). While $o_2$ is used to determine whether the event is governed by the cue, and Non-Uu events are classified into CT+/-, PR+/-, PS+/- according to $o_2$. The dimension of $o_2$ is 3 (label $y_2$), considering that a sentence may have no cues. The objective function with $l_2$-norm is designed as:

$$J(\theta) = \epsilon[-\frac{1}{m}\sum_{i=0}^{m-1}\log p(y_1^{(i)}|x^{(i)},\theta)]+ \quad (23)$$

$$(1-\epsilon)[-\frac{1}{m}\sum_{i=0}^{m-1}\log p(y_2^{(i)}|x^{(i)},\theta)] + \frac{\lambda}{2}\|\theta\|^2$$

where given the training instance $(x, y_1)$ and $(x, y_2)$, $p(y_1^{(i)}|x^{(i)}, \theta)$ and $p(y_2^{(i)}|x^{(i)}, \theta)$ are the confidence scores of the golden label $y_1$ and $y_2$ in $o_1$ and $o_2$, respectively.

## 5 Experimentation

This section first introduces experimental settings and then gives the detailed results and analysis of event factuality identification.

### 5.1 Experimental Settings

We evaluate our models on FactBank (Saurí and Pustejovsky, 2009), which contains 3864 sentences, 9492 events and 13506 factuality values. Table 2 presents the distribution of factuality values in FactBank. Following previous studies (Saurí, 2008; de Marneffe et al., 2012), we only take into account the five main categories of values, i.e., CT+, CT-, PR+, PS+, and Uu, which make up 99.05% of all the instances.

|       | Counts | Percentage(%) |
|-------|--------|---------------|
| CT+   | 7749   | 53.37         |
| CT-   | 433    | 3.21          |
| PR+   | 363    | 2.69          |
| PS+   | 226    | 1.67          |
| Uu    | 4607   | 34.11         |
| Other | 128    | 0.95          |
| Total | 13506  | 100           |

Table 2: Distribution of factuality values in Fact-Bank.

We divide the corpus into 5 folds to perform 5-fold cross-validation. Precision, Recall, and F1-measure are employed to report the performance of each factuality value. To obtain the performance for the whole corpus, macro-averaging and micro-averaging are applied (Manning and Schütze, 2001). We choose *two-sample two-tailed t-test* for significance test.

For the SIP detection task, we set $d_0 = 100$, $d_{pos} = d_{hyp} = 50$, $n_c = 150$. For the event factuality identification task, we set $d_0 = 100$, $d_p = d_t = 10$, $n_c = 50$, $\epsilon = 0.25$. We initialize word embeddings via Word2Vec (Mikolov et al., 2013) and randomly initialize all the other parameters. The Stochastic Gradient Descent with momentum is applied to optimize our models.

If there is more than one cue in the sentence, we consider whether the event is modified by each cue separately. If a Non-Uu event is affected by both PR and PS cues, we adopt the cue with the

highest confidence score in $o_2$ to decide whether the modality of the event is PR or PS.

We employ the following baselines, whose inputs are the outputs of our basic factor extraction for the fair comparison with our model:

**Rules**: The rule-based model developed by Saurí (2008) is a top-down algorithm traversing a dependency parse tree.

**MaxEnt**: We employ a maximum entropy classification model using the features developed by de Marneffe et al. (2012). Some features are not available (i.e., *predicate classes* and *general classes of events*) because they rely on annotated information which are not outputs of our basic factor extraction tasks.

**CNN+CNN**: This is a variant of our model whose LSTM is replaced by the CNN in Section 3 with the size of convolutional layer $n_c = 100$.

### 5.2 Results and Analysis: Basic Factor Extraction

Table 3 presents the performances of basic factor extraction tasks. It is worth noting that a SIP is correctly identified means both this SIP and the new source introduced by it are correctly detected.

|       |     | P(%)  | R(%)  | F     |
|-------|-----|-------|-------|-------|
| Event |     | 86.67 | 82.86 | 84.68 |
| SIP   |     | 73.98 | 73.01 | 73.42 |
| Source of events | | 79.30 | 76.53 | 77.82 |
| Cue   | All | 66.52 | 65.68 | 66.04 |
|       | NEG | 72.13 | 75.13 | 73.48 |
|       | PR  | 55.92 | 47.97 | 51.36 |
|       | PS  | 62.41 | 65.90 | 63.47 |

Table 3: The performances of basic factor extraction tasks.

We get a better performance on event detection task (F1=84.68) than Chambers (2013) (F1=80.30). Remember that we only consider nouns, verbs, adjectives, and rule out the effects of other tokens. We also employ the features developed by Chambers (2013) to identify SIPs and obtain F1=72.56, while our attention-based CNN for SIP detection achieves a higher F1=73.42, proving that our CNN is beneficial from pruned sentences (PSen) structures. Although the improvement of F1 is slight, however, we argue that one SIP can determine all the sources of events embedded in it. Therefore, we adopt the attention-based CNN model for SIP detection.

| Systems | Sources | CT+ | CT- | PR+ | PS+ | Uu | Micro-A | Macro-A |
|---|---|---|---|---|---|---|---|---|
| Rules | All | 61.83 | 54.52 | 20.75 | 39.89 | 26.08 | 50.71 | 40.62 |
| | Author | 64.83 | 48.83 | 13.35 | 31.93 | 26.17 | 53.49 | 37.02 |
| | Embed | 53.19 | 61.94 | 25.11 | 47.01 | 25.95 | 44.20 | 42.64 |
| MaxEnt | All | 64.37 | 47.57 | 35.74 | 17.08 | 58.86 | 61.00 | 44.87 |
| | Author | 70.93 | 48.36 | 28.25 | 14.67 | 67.19 | 68.41 | 50.30 |
| | Embed | 50.61 | 45.78 | 38.49 | 24.97 | 24.26 | 43.69 | 36.82 |
| CNN+CNN(Avg) | All | 60.15 | 51.53 | 32.88 | 34.06 | 56.67 | 57.66 | 47.06 |
| | Author | 66.11 | 54.67 | 29.25 | 34.67 | 63.44 | 64.14 | 49.69 |
| | Embed | 47.88 | 48.35 | 34.92 | 36.49 | 27.09 | 42.59 | 38.95 |
| CNN+CNN(Att) | All | 62.50 | 52.56 | 42.56 | 40.02 | 56.69 | 59.43 | 50.87 |
| | Author | 68.97 | 53.86 | 39.26 | 35.91 | 64.16 | 66.23 | 52.43 |
| | Embed | 49.37 | 50.76 | 43.57 | 41.10 | 25.32 | 43.69 | 42.03 |
| BiLSTM+CNN(Avg) | All | 61.74 | 49.19 | 34.09 | 40.19 | 55.59 | 58.23 | 48.16 |
| | Author | 67.67 | 50.33 | 34.55 | 31.10 | 62.94 | 64.79 | 49.32 |
| | Embed | 49.68 | 47.95 | 34.48 | 43.13 | 22.86 | 42.94 | 39.62 |
| BiLSTM+CNN(Att) | All | 64.17 | 53.89 | 41.83 | 42.97 | 58.46 | 61.06 | 52.26 |
| | Author | 70.16 | 55.39 | 36.84 | 31.75 | 65.90 | 67.59 | 52.01 |
| | Embed | 51.68 | 52.51 | 43.39 | 47.45 | 26.26 | 45.81 | 44.26 |

Table 4: The performances (F1-measures) of all the systems on event factuality identification. "Embed" are the sources in the chain form embedded in *AUTHOR*. "Avg" means using average pooling layers, and "Att" denotes the attention mechanism.

## 5.3 Results and Analysis: Models

Table 4 displays the performances of various models on event factuality identification. CT-, PR+ and PS+ only cover 7.57% of all the factuality values. Hence, it is challenging to identify them. Compared with Rule-based and MaxEnt model, our attention-based neural network models achieve better results on CT-, PR+ and PS+. The higher macro-averaging indicates the performances are more balanced, and BiLSTM+CNN(Att) is superior to all the other baselines in terms of micro- and macro-averaging on All sources. However, if we concatenate $f_u$ and $f_{cue}$ in Equation (19)(20) into ONE vector and design only ONE output in BiLSTM+CNN(Att), we obtain good results on CT+ and Uu (F1=63.64 and 57.80), but results of CT-, PR+, PS+ are lower (F1=49.43, 33.06, 37.07). With the design of two outputs, BiLSTM+CNN(Att) achieves the highest F1 of PS+ (42.97) and the second highest F1 of CT-, PR+ (53.89 and 41.83), which can prove the advantages of the design of two outputs in our model and the usefulness of speculative and negative cues.

Particularly, our BiLSTM model can obtain better results than CNN. Compared with the simple CNN which ignores the contexts, BiLSTM can learn features from both future and past contexts. Hence BiLSTM is able to ex-

tract more effective representations from syntactic paths than CNN. For BiLSTM+CNN(Att) and CNN+CNN(Att) models, we get $p < 0.05$.

The performances of both CNN+CNN ($p < 0.001$) and BiLSTM+CNN ($p < 0.05$) models are improved after using attention, indicating that some factors (e.g., SIPs, cues, auxiliary words) in syntactic paths play the key roles in computing the factuality, while others have fewer effects. Therefore, attention is helpful to improve the results.

The F1 of BiLSTM+CNN(Att) on CT+ and Uu events whose sources are *AUTHOR* are slightly lower than those of MaxEnt model. CT+ and Uu values cover 95.45% among events with *AUTHOR*, so MaxEnt can also perform well on identifying them. But BiLSTM+CNN(Att) achieves higher F1 on CT- ($p < 0.001$), PR+ ($p < 0.05$) and PS+ ($p < 0.001$) on *AUTHOR*.

However, F1 of Uu on embedded sources of BiLSTM+CNN(Att) is quite low (26.26), which is similar to other models. On one hand, Uu only covers 23.99% among events with embedded sources. On the other hand, the events embedded in other sources usually have complex syntactic structures. Therefore, it is difficult to judge whether these events are Uu or Non-Uu. Compared with MaxEnt model, however, BiLSTM+CNN(Att) achieves better performances on

CT-, PR+, PS+ events with *AUTHOR* (CT-, PS+: $p < 0.001$) and on all the five categories of values with embedded sources ($p < 0.001$). These results can prove the effectiveness of our model, especially cue-related features which are helpful to identify speculative and negative values.

## 5.4 Results and Analysis: Effects of Inputs

We explore the effects of different inputs on the performances of our BiLSTM+CNN(Att) model and obtain the results in Figure 6. We can conclude that only RS_Path is not sufficient to identify Uu and Non-Uu events. SIP_Path can greatly improve the results ($p < 0.001$), proving that SIPs are key factors to determine the event factuality and can offer meaningful syntactic information to discriminate Non-Uu events from Uu ones. Finally, auxiliary words make further improvement on results, because they can reflect the pragmatic structures of sentences, just as described in Section 4. For example:

(S6) *He also is* **charged** *in the 1996 Olympic* **bombing**, *which* **killed** *one person.*

(S7) *The environmental commission must adopt regulations to* **ensure** *people are not exposed to radioactive waste.*

In S6, the auxiliary word "*which*" leads the relative clause "*which* **killed** *one person*" containing the CT+ event **killed**. Events in non-restrictive clauses are commonly evaluated as facts. In S7, event **ensure** is in the purpose clause and is presented as Uu grammatically.

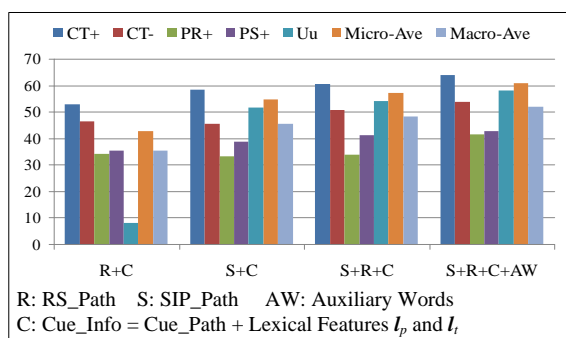

Figure 6: F1-measures of our BiLSTM+CNN(Att) model using different inputs.

## 6 Related Work

Currently, only a few studies focused on event factuality identification. Diab et al. (2009) and Prabhakaran et al. (2010) presented a preliminary pilot study of belief annotation and automatic tagging,

and limited the source to *AUTHOR*. They classified predicates into Committed Belief (CB), Non-CB or Not Applicable under a supervised framework using lexical and syntactic features.

Saurí (2008) proposed a rule-based top-down algorithm traversing dependency trees which can achieve excellent results (F1 of macro- and micro-averaging are 73.59 and 85.45, respectively), and developed FactBank corpus (Saurí and Pustejovsky, 2009). Then de Marneffe et al. (2012) employed lexical, syntactic and pragmatic features to identify factuality of events in some sentences of FactBank. These studies utilized annotated events and relative factors directly.

Neural networks, emerging in recent years, can not only extract effective features from sentences (Socher et al., 2012; Zeng et al., 2014; Zhou and Xu, 2015), but also capture important information from syntactic paths and improve performances of many NLP applications, such as relation classification (Xu et al., 2015a,b) and semantic role labeling (Roth and Lapata, 2016).

Many recent researches show that attention-based models can improve the performances of NLP tasks. Chen et al. (2016) built a hierarchical LSTM model with attention to generate sentence and document representations for sentiment classification. Zhou et al. (2016) utilized a bidirectional LSTM with attention to model sentences, and Wang et al. (2016) proposed a novel CNN relying on two levels of attention in terms of relation classification. In this paper we utilize attention-based neural networks which consider both LSTM and CNN to learn effective features from syntactic paths and words, respectively.

## 7 Conclusions

This paper presents a supervised framework to identify factuality of events from raw texts. We firstly extract events, SIPs, relevant sources and cues in texts, and then employ an attention-based neural network model combining BiLSTM and CNN for event factuality identification. Our model can identify negative and speculative factuality values more effectively with the help of corresponding cues. Experimental results show that our model can outperform the state-of-the-art ones. We will develop more useful features for basic factor extraction tasks and design better neural network models to improve the results of event factuality identification in the future work.

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
