# Peer review of "Event Factuality Identification via Deep Neural Networks"

_ACL 2017 — decision unknown_

[Official Review · Reviewer 1 · rating 3 · confidence 4]
soundness 5 · originality 3 · clarity 4 · impact 3 · substance 3 · appropriateness 5 · meaningful comparison 5 · presentation format Poster

Update after author response: 

1. My major concern about the optimization of model's hyperparameter (which are
numerous) has not been addressed. This is very important, considering that you
report results from folded cross-validation. 

2. The explanation that benefits of their method are experimentally confirmed
with 2% difference -- while evaluating via 5-fold CV on 200 examples -- is
quite unconvincing.

========================================================================

Summary:

In this paper authors present a complex neural model for detecting factuality
of event mentions in text. The authors combine the following in their complex
model:                          (1) a set of traditional classifiers for detecting
event
mentions,
factuality sources, and source introducing predicates (SIPs), (2) A
bidirectional attention-based LSTM model that learns latent representations for
elements on different dependency paths used as input, (2) A CNN that uses
representations from the LSTM and performs two output predictions (one to
detect specific from underspecified cases and another to predict the actual
factuality class). 

From the methodological point of view, the authors are combining a reasonably
familiar methods (att-BiLSTM and CNN) into a fairly complex model. However,
this model does not take raw text (sequence of word embeddings) as input, but
rather hand-crafted features (e.g., different dependency paths combining
factuality concepts, e.g., sources, SIPs, and clues). The usage of hand-crafted
features is somewhat surprising if coupled with complex deep model. The
evaluation seems a bit tainted as the authors report the results from folded
cross-validation but do not report how they optimized the hyperparameters of
the model. Finally, the results are not too convincing -- considering the
complexity of the model and the amount of preprocessing required (extraction of
event mentions, SIPs, and clues), a 2% macro-average gain over the rule-based
baseline and overall 44% performance seems modest, at best (looking at
Micro-average, the proposed model doesn't outperform simple MaxEnt classifier).

The paper is generally well-written and fairly easy to understand. Altogether,
I find this paper to be informative to an extent, but in it's current form not
a great read for a top-tier conference.   

Remarks:

1. You keep mentioning that the LSTM and CNN in your model are combined
"properly" -- what does that actually mean? How does this "properness"
manifest? What would be the improper way to combine the models?

2. I find the motivation/justification for the two output design rather weak: 
    - the first argument that it allows for later addition of cues (i.e
manually-designed features) kind of beats the "learning representations"
advantage of using deep models. 
        - the second argument about this design tackling the imbalance in the
training set is kind of hand-wavy as there is no experimental support for this
claim. 

3. You first motivate the usage of your complex DL architecture with learning
latent representations and avoiding manual design and feature computation.  And
then you define a set of manually designed features (several dependency paths
and lexical features) as input for the model. Do you notice the discrepancy? 

4. The LSTMs (bidirectional, and also with attention) have by now already
become a standard model for various NLP tasks. Thus I find the detailed
description of the attention-based bidirectional LSTM unnecessary. 
5. What you present as a baseline in Section 3 is also part of your model (as
it generates input to your model). Thus, I think that calling it a baseline
undermines the understandability of the paper. 

6. The results reported originate from a 5-fold CV. However, the model contains
numerous hyperparameters that need to be optimized (e.g., number of filters and
filter sizes for CNNs). How do you optimize these values? Reporting results
from a folded cross-validation doesn't allow for a fair optimization of the
hypeparameters: either you're not optimizing the model's hyperparameters at
all, or you're optimizing their values on the test set (which is unfair). 

7. "Notice that some values are non-application (NA) grammatically, e.g., PRu,
PSu, U+/-" -- why is underspecification in ony one dimension (polarity or
certainty) not an option? I can easily think of a case where it is clear the
event is negative, but it is not specified whether the absence of an event is
certain, probable, or possible. 

Language & style:

1. "to a great degree" -> "great degree" is an unusual construct, use either
"great extent" or "large degree"
2. "events that can not" -> "cannot" or "do not"
3. "describes out networks...in details shown in Figure 3." -> "...shown in
Figure 3 in details."

[Official Review · Reviewer 2 · rating 3 · confidence 5]
soundness 5 · originality 3 · clarity 5 · impact 3 · substance 3 · appropriateness 5 · meaningful comparison 5 · presentation format Poster

Comments after author response

- Thank you for clarifying that the unclear "two-step framework" reference was
not about the two facets. I still do not find this use of a pipeline to be a
particularly interesting contribution.
- You state that "5. de Marneffe (2012) used additional annotated features in
their system. For fair comparison, we re-implement their system with annotated
information in FactBank." But the de Marneffe et al. feature cited in the
paper, "Predicate Classes" requires only a dependency parser and vocabulary
lists from Roser Saurí's PhD thesis; "general classes of event" might be
referring to FactML event classes, and while I admit it is not particularly
clear in their work, I am sure they could clarify.
- I continue to find the use of "combined properly" to be obscure. I agree that
using LSTM and CNN where respectively appropriate is valuable, but you seem to
imply that some prior work has been improper, and that it is their combination
which must be proper.
- Thank you for reporting on separate LSTMs for each of the paths. I am curious
as to why this combination may less effective. In any case, experiments with
this kind of alternative structure deserve to be reported.

---

This paper introduces deep neural net technologies to the task of factuality
classification as defined by FactBank, with performance exceeding alternative
neural net models and baselines reimplemented from the literature.

- Strengths:

This paper is very clear in its presentation of a sophisticated model for
factuality classification and of its evaluation.  It shows that the use of
attentional features and BiLSTM clearly provide benefit over alternative
pooling strategies, and that the model also exceeds the performance of a more
traditional feature-based log-linear model.  Given the small amount of training
data in FactBank, this kind of highly-engineered model seems appropriate. It is
interesting to see that the BiLSTM/CNN model is able to provide benefit despite
little training data.

- Weaknesses:

My main concerns with this work regard its (a) apparent departure from the
evaluation procedure in the prior literature; (b) failure to present prior work
as a strong baseline; and (c) novelty.

While I feel that the work is original in engineering deep neural nets for the
factuality classification task, and that such work is valuable, its approach is
not particularly novel, and "the proposal of a two-step supervised framework"
(line 087) is not particularly interesting given that FactBank was always
described in terms of two facets (assuming I am correct to interpret "two-step"
as referring to these facets, which I may not be).

The work cites Saurí and Pustejovsky (2012), but presents their much earlier
(2008) and weaker system as a baseline; nor does it consider Qian et al.'s
(IALP 2015) work which compares to the former.              Both these works are
developed
on the TimeBank portion of FactBank and evaluated on a held-out ACQUAINT
TimeBank section, while the present work does not report results on a held-out
set.

de Marneffe et al.'s (2012) system is also chosen as a baseline, but not all
their features are implemented, nor is the present system evaluated on their
PragBank corpus (or other alternative representations of factuality proposed in
Prabhakaran et al. (*SEM 2015) and Lee et al. (EMNLP 2015)).  The evaluation is
therefore somewhat lacking in comparability to prior work.

There were also important questions left unanswered in evaluation, such as the
effect of using gold standard events or SIPs.

Given the famed success of BiLSTMs with little feature engineering, it is
somewhat disappointing that this work does not attempt to consider a more
minimal system employing deep neural nets on this task with, for instance, only
the dependency path from a candidate event to its SIP plus a bag of modifiers
to that path. The inclusion of heterogeneous information in one BiLSTM was an
interesting feature, which deserved more experimentation: what if the order of
inputs were permuted? what if delimiters were used in concatenating the
dependency paths in RS instead of the strange second "nsubj" in the RS chain of
line 456? What if each of SIP_path, RS_path, Cue_path were input to a separate
LSTM and combined? The attentional features were evaluated together for the CNN
and BiLSTM components, but it might be worth reporting whether it was
beneficial for each of these components. Could you benefit from providing path
information for the aux words? Could you benefit from character-level
embeddings to account for morphology's impact on factuality via tense/aspect?
Proposed future work is lacking in specificity seeing as there are many
questions raised by this model and a number of related tasks to consider
applying it to.

- General Discussion:

194: Into what classes are you classifying events?

280: Please state which are parameters of the model.

321: What do you mean by "properly"? You use the same term in 092 and it's not
clear which work you consider improper nor why.

353: Is "the chain form" defined anywhere? Citation? The repetition of nsubj in
the example of line 456 seems an unusual feature for the LSTM to learn.

356: It may be worth footnoting here that each cue is classified separately.

359: "distance" -> "surface distance"

514: How many SIPs? Cues? Perhaps add to Table 3.

Table 2. Would be good if augmented by the counts for embedded and author
events. Percentages can be removed if necessary.

532: Why 5-fold? Given the small amount of training data, surely 10-fold would
be more useful and not substantially increase training costs.

594: It's not clear that this benefit comes from PSen, nor that the increase is
significant or substantial.  Does it affect overall results substantially?

674: Is this significance across all metrics?

683: Is the drop of F1 due to precision, recall or both?

686: Not clear what this sentence is trying to say.

Table 4: From the corpus sizes, it seems you should only report 2 significant
figures for most columns (except CT+, Uu and Micro-A).

711: It seems unsurprising that RS_path is insufficient given that the task is
with respect to a SIP and other inputs do not encode that information. It would
be more interesting to see performance of SIP_path alone.

761: This claim is not precise, to my understanding. de Marneffe et al (2012)
evaluates on PragBank, not FactBank.

Minor issues in English usage:

112: "non-application" -> "not applicable"

145: I think you mean "relevant" -> "relative"

154: "can be displayed by a simple source" is unclear

166: Not sure what you mean by "basline". Do you mean "pipeline"?

[Official Review · Reviewer 3 · rating 2 · confidence 4]
soundness 5 · originality 3 · clarity 2 · impact 3 · substance 2 · appropriateness 5 · meaningful comparison 5 · presentation format Poster

This paper proposes a supervised deep learning model for event factuality
identification.  The empirical results show that the model outperforms
state-of-the-art systems on the FactBank corpus, particularly in three classes
(CT-, PR+ and PS+).  The main contribution of the paper is the proposal of an
attention-based two-step deep neural model for event factuality identification
using bidirectional long short-term memory (BiLSTM) and convolutional neural
network (CNN).

[Strengths:]

- The structure of the paper is (not perfectly but) well organized.

- The empirical results show convincing (statistically significant) performance
gains of the proposed model over strong baseline.

[Weaknesses:]

See below for details of the following weaknesses:

- Novelties of the paper are relatively unclear.

- No detailed error analysis is provided.

- A feature comparison with prior work is shallow, missing two relevant papers.

- The paper has several obscure descriptions, including typos.

[General Discussion:]

The paper would be more impactful if it states novelties more explicitly.  Is
the paper presenting the first neural network based approach for event
factuality identification?  If this is the case, please state that.

The paper would crystallize remaining challenges in event factuality
identification and facilitate future research better if it provides detailed
error analysis regarding the results of Table 3 and 4.              What are dominant
sources of errors made by the best system BiLSTM+CNN(Att)?  What impacts do
errors in basic factor extraction (Table 3) have on the overall performance of
factuality identification (Table 4)?  The analysis presented in Section 5.4 is
more like a feature ablation study to show how useful some additional features
are.

The paper would be stronger if it compares with prior work in terms of
features.  Does the paper use any new features which have not been explored
before?  In other words, it is unclear whether main advantages of the proposed
system come purely from deep learning, or from a combination of neural networks
and some new unexplored features.  As for feature comparison, the paper is
missing two relevant papers:

- Kenton Lee, Yoav Artzi, Yejin Choi and Luke Zettlemoyer. 2015 Event Detection
and Factuality Assessment with Non-Expert Supervision. In Proceedings of the
2015 Conference on Empirical Methods in Natural Language Processing, pages
1643-1648.

- Sandeep Soni, Tanushree Mitra, Eric Gilbert and Jacob Eisenstein. 2014.
Modeling Factuality Judgments in Social Media Text. In Proceedings of the 52nd
Annual Meeting of the Association for Computational Linguistics, pages 415-420.

The paper would be more understandable if more examples are given to illustrate
the underspecified modality (U) and the underspecified polarity (u).  There are
two reasons for that.  First, the definition of 'underspecified' is relatively
unintuitive as compared to other classes such as 'probable' or 'positive'. 
Second, the examples would be more helpful to understand the difficulties of Uu
detection reported in line 690-697.  Among the seven examples (S1-S7), only S7
corresponds to Uu, and its explanation is quite limited to illustrate the
difficulties.

A minor comment is that the paper has several obscure descriptions, including
typos, as shown below:

- The explanations for features in Section 3.2 are somewhat intertwined and
thus confusing.  The section would be more coherently organized with more
separate paragraphs dedicated to each of lexical features and sentence-level
features, by:

  - (1) stating that the SIP feature comprises two features (i.e.,
lexical-level
and sentence-level) and introduce their corresponding variables (l and c) *at
the beginning*;

  - (2) moving the description of embeddings of the lexical feature in line
280-283
to the first paragraph; and

  - (3) presenting the last paragraph about relevant source identification in a
separate subsection because it is not about SIP detection.

- The title of Section 3 ('Baseline') is misleading.  A more understandable
title would be 'Basic Factor Extraction' or 'Basic Feature Extraction', because
the section is about how to extract basic factors (features), not about a
baseline end-to-end system for event factuality identification.

- The presented neural network architectures would be more convincing if it
describes how beneficial the attention mechanism is to the task.

- Table 2 seems to show factuality statistics only for all sources.  The table
would be more informative along with Table 4 if it also shows factuality
statistics for 'Author' and 'Embed'.

- Table 4 would be more effective if the highest system performance with
respect to each combination of the source and the factuality value is shown in
boldface.

- Section 4.1 says, "Aux_Words can describe the *syntactic* structures of
sentences," whereas section 5.4 says, "they (auxiliary words) can reflect the
*pragmatic* structures of sentences."  These two claims do not consort with
each other well, and neither of them seems adequate to summarize how useful the
dependency relations 'aux' and 'mark' are for the task.

- S7 seems to be another example to support the effectiveness of auxiliary
words, but the explanation for S7 is thin, as compared to the one for S6.  What
is the auxiliary word for 'ensure' in S7?

- Line 162: 'event go in S1' should be 'event go in S2'.

- Line 315: 'in details' should be 'in detail'.

- Line 719: 'in Section 4' should be 'in Section 4.1' to make it more specific.

- Line 771: 'recent researches' should be 'recent research' or 'recent
studies'.  'Research' is an uncountable noun.

- Line 903: 'Factbank' should be 'FactBank'.